# INHIBITION OF THE PROSTAGLANDIN-DEGRADING ENZYME 15-PGDH AMELIORATES MASH-ASSOCIATED APOPTOSIS AND FIBROSIS IN MICE

**DOI:** 10.3390/cells14130987

**Published:** 2025-06-27

**Authors:** Utibe-Abasi S. Udoh, Mathew Steven Schade, Jacqueline A. Sanabria, Pradeep Kumar Rajan, Rodrigo Aguilar, Micheal Andryka, Alexei Gorka, Sandrine V. Pierre, Juan Sanabria

**Affiliations:** 1Department of Surgery, Marshall University Joan Edwards School of Medicine, Huntington, WV 25701, USA; udohu@marshall.edu (U.-A.S.U.); schade4@live.marshall.edu (M.S.S.); sanabriaja@marshall.edu (J.A.S.); rajan@marshall.edu (P.K.R.); aguilarcampo@marshall.edu (R.A.); andrykamb@gmail.com (M.A.); asgorka@gmail.com (A.G.); 2Marshall Institute for Interdisciplinary Research (MIIR), Marshall University, Huntington, WV 25701, USA; pierres@marshall.edu; 3Department of Nutrition and Metabolomic Core Facility, School of Medicine, Case Western Reserve University, Cleveland, OH 44100, USA

**Keywords:** glutathione, prostaglandin inhibitor, metabolism, MAFLD, MASH, liver fibrosis

## Abstract

*Background*. Metabolic dysfunction-associated steatotic liver disease (MASLD) affects more than 30% of the world population. Progression to its inflammatory state, MASH, is associated with increasing liver fibrosis, leading to end-stage liver disease (ESLD) and hepatocellular carcinoma (HCC). SW033291, an inhibitor of 15-PGDH (the PGE2 degradation enzyme), has been shown to increase in vivo regeneration of liver parenchyma, ameliorating oxidative stress and inflammation. We hypothesized that SW033291 abrogates MASH progression by inducing a paucity of the initial apoptotic switch and restoring physiological collagen’s microenvironment. *Methods*. The expression levels of the cell metabolic proteins FOXO1, mTOR, and SIRT7 were determined in a diet-induced MASH-mouse model at 16, 20, and 24 weeks. Non-targeted metabolomics in mouse plasma were measured by LC-MS/MS. Liver morphology and apoptotic activity were quantified by the NAS score and TUNEL assay, respectively. Statistical analyses between groups (NMC, HFD, and SW033291) were determined by ANOVA, *t*-test/Tukey’s post hoc test using GraphPad Prism. Metabolomics data were analyzed using R-lab. *Results*. The treated group showed significant decreases in total body fat, cellular oxidative stress, and inflammation and an increase in total lean mass with improved insulin resistance and favorable modulation of metabolic protein expressions (*p* < 0.05). SW033291 significantly decreased GS:SG, citric acid, and corticosterone, NAS scores (9.4 ± 0.2 vs. 6.2 ± 0.1, *p* < 0.05), liver fibrosis scores (1.3 ± 0.5 vs. 0.25 ± 0.1, *p* < 0.05), and apoptotic activity (43.9 ± 4.6 vs. 0.38 ± 0.1%, *p* < 0.05) compared with controls at 24W. *Conclusions*. The inhibition of 15-PGDH appears to normalize the metabolic and morphological disturbances during MASH progression with a paucity of the initial apoptotic switch, restoring normal collagen architecture. SW033291 warrants further investigation for its translation.

## 1. INTRODUCTION

Chronic liver disease accounts for nearly two million deaths/year, or 4% of all deaths globally [1]. Its morbidity in healthcare cost has an estimate of USD 20 B, representing a major financial burden and global health challenge [1,2,3]. A rising cause of chronic liver disease is metabolic dysfunction-associated steatotic liver disease (MASLD), which affects over 30% of the adult population worldwide [4]. MASLD can progress to metabolic dysfunction-associated steatohepatitis (MASH), a chronic and persistent inflammation state leading to end-stage liver disease (ESLD) [5,6,7]. MASH progression is often observed in the background of insulin resistance, central obesity, hypertension (HTN), and dyslipidemia [8], where parenchymal cell injury leads to hepatocellular damage and fibrosis [9,10,11]. At present, MASH is the leading cause of liver transplantation in the U.S., [12] rising as the most common cause of HCC [6,13]. The prevalence of MASH and its sequelae, ESLD and HCC, is related to the obesity epidemic, which affected 2 B people worldwide in 2015 [14] and is projected to affect nearly half of the U.S. adult population by 2030 [15].

The molecular pathways underlying MASH progression remain elusive. Nevertheless, it has been associated with cell oxidative stress, insulin resistance, metabolic dysfunction, and inflammation [5,6,7,8,9,10,11,16,17,18]. MASH manifests morphologically as macro/micro-steatosis, with advancing fibrosis and parenchymal cell apoptotic activity [5,6,7,8,9,10,11,16,17]. While dietary changes and exercise mitigate MASH progression, patient compliance remains challenging, and available treatment options are limited, associated with off-target effects and paucity of long-term efficacy [16,19]. Resmetirom (Rezdiffra®) is the only drug available for the treatment of MASH, which received accelerated FDA approval (March 2024) [20,21]. The degradation of Prostaglandin E2 (PGE2), a lipid signaling molecule involved in the regulation of inflammation and tissue stem cell activation, is catalyzed by the enzyme 15-hydroxyprostaglandin dehydrogenase (15-PGDH) [22,23,24,25,26]. SW033291 is a small-molecule sulfoxide (412.59 Da, Figure 1) and a potent inhibitor of 15-PGDH [27]. It has demonstrated the ability to increase levels of PGE2 in vivo [25]. We and others have shown in preclinical models that SW033291 promotes liver regeneration and ameliorates oxidative stress and inflammation [25,28,29]. In addition, it alleviated abnormal lipid metabolism, endoplasmic reticulum stress, and inflammatory responses [30]. SW033291 has also been shown to reduce cardiac fibrosis and alleviate diastolic dysfunction [31], exerting anti-fibrotic properties in idiopathic pulmonary fibrosis (IPF) [32]. Most recently, the inhibition of 15-PGDH attenuated acetaminophen-induced liver injury by suppressing apoptosis in liver endothelial cells [33]. In addition, SW033291 reduced liver necrosis and heightened GSH levels, with a concomitant decrease in tissue malondialdehyde (MDA) and the expression levels of inflammatory cytokines in a mouse model of acute liver injury [34]. Nevertheless, there is a gap in current knowledge on the effect of 15-PGDH inhibition on MASH progression. We hypothesized that SW033291 abrogates MASH progression through a paucity of the initial apoptotic switch, restoring physiological collagen’s microenvironment.

## 2. MATERIALS and METHODS

### 2.1. Animal Model

Seven-week-old female C57BL/6J mice were purchased from Jackson Laboratory (Farmington, CT) and housed in a 12:12 h light–dark cycle-controlled environment kept at 65–75°F (~18–23 °C) and 40–60% humidity. Mice were fed ad libitum with normal mouse chow (*NMC:* 7.2% fat and 61.6% carbohydrate, Bio-Serv #F4031, NJ) or a high-fat diet (*HFD*: 36.0% fat and 35.7% carbohydrate, Bio-Ser # F3282, NJ) supplemented with 55% fructose-in-water. After 12 weeks, the mice were assigned to the control (*NMC* and *HFD*) or experimental (*HFD + SW033291)* groups, *n* = 7/group, and maintained for another 4 (16 W), 8 (20 W), or 12 weeks (24 W), before concluding the study (Appendix A). At a power of 0.80, with a significance level established at <0.05, the sample size required to detect a response to cell apoptotic activity (the driver of MASH progression at the mean Δ of 50%) is *n* > 4 per group. We included *n* = 7 in each group, considering an estimated 20% animal dropout. The following solutions, 50 mM SW033291 dissolved in 10% ethanol, 5% Cremophor EL (Castor oil polyoxyethylene ether), and 85% D5W (Dextrose 5% in water) at room temperature (RT), were administered by intraperitoneal injection (IP twice/daily) at 5 mg/kg as previously described [25]. All animal work followed University IACUC-approved protocols. 

### 2.2. Metabolic Compartment Analysis

The metabolic body composition of mice was assessed a week prior to the end of the study by EchoMRI (100H Body Composition Analyzer, Houston, TX, USA). Total body weight (TBW), total body water, lean mass, and fat mass were recorded.

### 2.3. Plasma and Tissue Collection

Under anesthesia (pentobarbital, 5 mg/kg TBW, IP), a laparotomy was performed. Blood was drawn from the infra-hepatic vena cava (IVC), followed by liver excision. Livers were washed with 0.9% normal saline (NS) at RT and divided in half, with one portion being flash-frozen in liquid nitrogen and stored at −80 °C for later use and the other fixed with 10% buffered formalin at 4 °C. Prior to anesthetic induction (30 min), animals were administered octanoate (10 mg/100 μL 0.9% NS, pH adjusted to 7.4 at RT). Octanoate is a fatty acid (FA) used as a surrogate for the measurement of liver mitochondrial lipid β-oxidation [35]. IP octanoate reaches the liver via the portal vein, where it is oxidized into ketone bodies or carried into systemic circulation. The liver lipid β-oxidative capacity was determined by measuring the octanoate/butyrate ratio in peripheral blood using LC-MS/MS.

### 2.4. Plasma Treatment and Mass Spectrometry

Glutathione species (reduced glutathione (GSH), oxidized glutathione (GS:SG)), and non-targeted metabolites in treated plasma were measured by LC-MS/MS. Plasma treatment and liquid chromatography–mass spectrometry (LC-MS/MS) methods are described in detail in the expanded methods of the Appendix A.

### 2.5. Protein Expression

The expression levels of proteins involved in cellular metabolism and oxidative stress response (FOXO1, mTOR, and SIRT7) were evaluated using Western blot (WB). Briefly, homogenized liver tissues in RIPA buffer (pH = 7.4) were cleared by centrifugation (14,000 rpm/15 min/4 °C), and the supernatants were separated by SDS-PAGE and transferred to Protran nitrocellulose membranes (Thermo Fisher Scientific, Waltham, MA, USA). The membranes were blocked with 4% milk and incubated with protein primary antibodies to be probed with the corresponding horseradish peroxidase (HRP)-conjugated secondary antibodies. The membranes were developed using the Pierce ECL kit (Thermo Fisher Scientific, Waltham, MA, USA) with the FluorChem M System (Minneapolis, MN, USA). The blots were then quantitated using ImageJ-Fiji software (NIH, Bethesda, MD, USA) that measures the integrated density of each of the blots after background subtraction and normalizing against the housekeeping protein (β-Actin or α-Tubulin). The details of the antibodies used and their dilutions are listed in Appendix A.

### 2.6. Morphology

Paraffin-embedded blocks from formalin-fixed liver tissue were processed for staining using standard protocols (Bridges Lab Protocols & Abcam IHC deparaffinization protocol, Cambridge, MA, USA). Briefly, tissue blocks were cut at a thickness of 5–6 microns, and slide sections were subjected to deparaffinization and rehydration using xylene and a graded ethanol series before the specific staining procedure. Hematoxylin and eosin (H &E) and Trichrome staining are detailed in the expanded methods of the Appendix A. Images of histologically blinded liver sections were captured using a Leica confocal fixed-stage microscope (LEICA DM6000 CFS, Leica Microsystems Inc., Deerfield, IL, USA) or an EVOS microscope. Five regions per slide/liver were captured. ImageJ-Fiji software (version 2.16.0) (NIH, Bethesda, MD, USA) was used to count the positive cells/total cells and expressed as a percentage of cells or to measure the size of the cells. Data were analyzed using GraphPad Prism (9.5.1, licensed to the University).

### 2.7. Liver Fat Content

The NAS score was calculated using published criteria on macro-vesicular and micro-vesicular steatosis, inflammatory cell infiltrate, and cellular hypertrophy for each liver section under H&E staining [36,37]. Steatosis was graded as either macro (if the fat vacuoles displaced the nucleus) or micro (if there was no nuclear displacement) and scored as **0** = <5%, **1** = 5–33%, **2** = 34–66%, and **3** = >66%. Inflammatory foci were defined as an aggregate of more than 5 inflammatory cells and were scored as **0** (<0.5 foci), **1** (0.5–1.0 foci), **2** (1.0–2.0 foci), and **3** (>2.0 foci). Last, hepatocellular hypertrophy was defined as cellular enlargement greater than 1.5 times the average diameter of the hepatocyte’s diameter measured in the control slides [36,37]. Scoring was made blindly by two independent researchers. Five representative images at 40× magnification were captured for each liver section in all groups.

### 2.8. Liver Fibrosis

Sections from each mouse liver/group were stained with Masson’s Trichrome to assess collagen deposition. The whole liver section-stained images were graded according to the following scale, and aggregate scores were saved for analysis: 0: None; 1: Enlarged, fibrotic portal tracts; 2: Peri-portal or portal-portal septa but intact architecture; 3: Fibrosis with architectural distortion but no obvious cirrhosis; 4: Probable or definitive cirrhosis with bridging fibrosis [38,39]. Five representative images at 40× magnification were taken of the stained liver slides from each mouse liver in all groups. Grading was performed by two independent researchers and blind to the groups.

### 2.9. Liver Cellular Apoptosis

Apoptotic activity was detected by terminal deoxynucleotidyl transferase dUTP nick-end labeling (TUNEL) in liver tissue slides. A Click-iT Plus TUNEL assay kit (Invitrogen by Thermo Fisher Scientific, Waltham, MA, USA) was used for the assay in accordance with the manufacturer’s instructions. Five images at 40× magnification were taken of stained slides for each animal in every group. Positive cells/total cells were expressed as a percentage of apoptotic cells (ImageJ-Fiji software, NIH, Bethesda, MD, USA).

### 2.10. Statistical Analysis

The results are presented as box-and-whisker plots. Data are presented as the median (central line), first and third quartiles (bottom and top of boxes, respectively), and whiskers (extreme values) for each group of mice. Differences between groups were determined by analysis of variance (ANOVA), followed by a *t*-test or Tukey’s post hoc test using GraphPad Prism version 9.5.1 (GraphPad, San Diego, CA, USA). The statistical analysis of non-target metabolites was performed by R-lab (version 4.4.2), using the Kruskal–Wallis test, and data are presented as mean ± SD (bar charts). Non-parametric tests (Kruskal–Wallis) were used where possible to avoid assumptions of normality by comparing ranks across multiple groups rather than means. In addition, the Benjamini–Hochberg false discovery rate (FDR) correction over the set of individual metabolite tests. The Benjamini–Hochberg method was selected to control false discoveries without being overly conservative, ensuring that true biological differences could still be detected, given the moderate number of comparisons. In addition, PCA is an unsupervised method that captures overall variance without inferring causality; thus, it is inherently less sensitive to confounding than regression or univariate testing. Our analysis emphasizes group-level patterns across metabolites rather than the effects of individual metabolites. By applying PCA to a targeted set of biologically relevant metabolites that vary meaningfully between groups, we further minimized irrelevant variance and reduced the potential impact of confounding. Statistical significance was set at *p* < 0.05. Statistical tests, sample size, and *p*-values are provided in the figure legends.

## 3. RESULTS

### 3.1. Total Body Weight and Fat Mass

To assess the effects of SW033291 on TBW, fat mass, and metabolic body composition, Echo MRI measurements of the mice were taken a week prior to the end of this study. Our data revealed an increased TBW in both the HFD and SW033291 groups compared with the NMC group at weeks 16, 20, and 24 (TBW at 16 W: 33.6 ± 3.5 and 36.5 ± 1.2 vs. 24.1 ± 0.8 g., respectively; TBW at 20 W: 38.4 ± 2.3 and 40.6 ± 1.3 vs. 24.3 ± 0.8 g., respectively; TBW at 24 W: 39.9 ± 2.7 and 41.9 ± 1.4 vs. 24.6 ± 0.7 g., respectively, *p* < 0.05, Figure 2a). The fat mass followed a similar pattern, except at week 24, when we observed a significant decrease in the SW033291-treated group compared with HFD (fat mass at 24 W: 15.3 ± 0.6 vs. 20.0 ± 1.4 g., respectively, *p* < 0.05, Figure 2b). In addition, there was a significant increase in the lean mass of the SW033291-treated group mice compared with their counterparts, the HFD and NMC groups, at the three timepoints (lean mass at 24 W: 21.2 ± 0.2 vs. 20.4 ± 0.3 and 20.5 ± 0.2 g., respectively, *p* < 0.05, Figure 2c). A significant increase in total body water was observed at weeks 20 and 24 in the SW033291-treated group compared with the HFD group (total body water at 24 W: 18.0 ± 0.2 vs. 17.3 ± 0.3 g. respectively, *p* < 0.05, Figure 2d). Our data showed that SW033291 mice treated on HFD had less fat mass and increased lean mass and total body water.

### 3.2. Oxidative Stress (Glutathione Species and Ophthalmic Acid)

We measured the plasma level of oxidative stress markers GSH, GS:SG, and Ophthalmic acid (OA), as MASH development and progression are associated with oxidative stress [40]. There was a statistically significant increase in GSH and GS:SG in the HFD vs. NMC group at 24 W. In contrast, a significant decrease in both GSH and GS:SG was observed in the SW033291-treated mice vs. the HFD at 24 W (GSH at 24 W: 40.7 ± 4.3 vs. 16.7 ± 1.4 and 24.6 ± 1.2 vs. 40.7 ± 4.3 mmol/L, respectively, *p* < 0.05, Figure 3a; GS:SG at 24 W: 8.9 ± 1.5 vs. 2.4 ± 0.3 and 4.0 ± 0.3 vs. 8.9 ± 1.63 mmol/L, respectively, *p* < 0.05, Figure 3b). Interestingly, we observed a significant reduction in the GSH to GS:SG ratio in the HFD group compared with the NMC group (GSH/GS:SG at 24 W: 4.95 ± 0.7 vs. 7.05 ± 0.3, respectively, *p* < 0.05, Figure 3c). Plasma Ophthalmic acid levels significantly increased in the HFD and SW033291 groups across the experimental weeks and groups when compared with the NMC group (OA level at 24 W: 0.47 ± 0.2 and 0.20 ± 0.1 vs. 0.01 ± 0.03 ng/mL; respectively, *p* < 0.05, Figure 3d) at 24 W. These results indicate that SW033291 improved the redox status of liver cells.

### 3.3. Insulin Resistance and Lipid β-Oxidation

To investigate the effects of SW033291 on insulin resistance and lipid β-oxidation, the blood glucose level and plasma octanoate/butyrate ratio (O:B) were measured at the end of each experimental timepoint. There was a significant increase in blood glucose (mmol/L) at week 24 in the HFD group compared with NMC. However, the SW033291 treatment led to a significant decrease toward normal values (blood glucose at 24 W: 365.9 ± 71.1 vs. 124.3 ± 41.4 and 207.8 ± 40.1 mmol/L, respectively, *p* < 0.05, Figure 4a). However, there was no significant difference in the O:B ratio among the three groups at weeks 16 and 20, but we observed a significant increase in the HFD group compared with NMC at week 24, which was not significantly different from the SW033291-treated mice (O:B at 24 W: 0.10 ± 0.1 vs. 0.00± 0.0 and 0.05 ± 0.0, respectively, *p* < 0.05, Figure 4b). Our findings suggest that SW033291 normalized blood glucose but has no effect on lipid β-oxidation.

### 3.4. Non-Targeted Metabolomics

Accumulating data indicate that metabolomics can assist in understating various metabolic and signaling pathways in liver health and diseases such as MASH [41]. There were significant variations in the plasma concentrations of GS:SG, citric acid, and corticosterone among groups at 24 W (NMC vs. HFD vs. SW033291-treated mice). GS:SG (19.6 ± 12.0 vs. 58.4 ± 28.2 vs. 25.9 ± 78.5 mmol/L, respectively, *p* < 0.05, Table 1a,d, Figure 5 and Appendix A), citric acid (2708.1 ± 201.4 vs. 3746.8 ± 460.4 vs. 2775.6 ± 669.8 mmol/L, respectively, *p* < 0.05, Table 1b,d, Figure 5 and Appendix A), and corticosterone (51.4 ± 5.7 vs. 85.0 ± 11.3 vs. 55.8 ± 21.3 mmol/L, respectively, *p* < 0.05, Table 1c,d, Figure 5 and Appendix A), were significantly higher in the HFD group when compared with the NMC group, with a significant improvement towards normalization in the SW033291-treated group. Overall, SW033291 significantly decreased plasma concentrations of GS:SG, citric acid, and corticosterone towards a normalization of the citric acid cycle.

### 3.5. Protein Expression

To determine the molecular targets behind changes in cellular redox status and altered metabolism in our mouse model, we assayed expressions of various key proteins that are involved in liver metabolic regulation. FOXO1 and mTOR expression were significantly lower in the SW033291-treated group vs. HFD along almost all study points (FoxO1 at 24 W: 0.16 ± 0.1 vs. 1.19 ± 0.6, respectively, *p* < 0.05, Figure 6a and mTOR at 24 W: 0.46 ± 0.1 vs. 0.95 ± 0.1, respectively, *p*< 0.05, Figure 6b). In addition, there was a significant decrease in SIRT7 expression in the SW033291-treated mice vs. HFD and NMC at 24 W (SIRT7 at 24 W: 0.45 ± 0.1 vs. 0.81 ± 0.1 and 1.00 ± 0.1 respectively, *p* < 0.05, Figure 6c). Our data indicate that SW033291 downregulated the expression of FOXO1, mTOR, and SIRT7.

### 3.6. Liver Morphology

#### 3.6.1. The NAS Score

The HFD group demonstrated a significant increase in NAS scores at 16, 20, and 24 W compared with NMC (Figure 7A,B(i) and Appendix A). The NAS score in the SW033291-treated mice initially increased compared with the NMC and HFD at 16 W, but at 20 and 24 W, this trend was reversed with a decrease in NAS score (NAS Score at 24 W: 3.7 ± 0.0 and 9.4 ± 0.2 vs. 6.2 ± 0.1, respectively, *p* < 0.05, Figure 7A,B(i) and Appendix A).

#### 3.6.2. Fibrosis Score

The HFD group had a significantly higher fibrosis score at 20 and 24 W compared with the NMC and SW033291 groups (Fibrosis Score at 24 W: 1.28 ± 0.5 vs. 0.00 ± 0.0 and 0.25 ± 0.1, respectively, *p* < 0.05, Figure 7A,B(ii) and Appendix A).

#### 3.6.3. Apoptotic Activity

The apoptotic activity was significantly higher in the HFD group when compared with both the NMC and SW033291 groups at 24 W (apoptosis at 24 W: 43.9 ± 4.6 vs. 0.38 ± 0.2 and 0.38 ± 0.1, respectively, *p* < 0.05, Figure 7A,B(iii) and Appendix A). Our results indicate that SW033291 regressed the morphological manifestations of MASH, restoring normal liver collagen disposition patterns and abrogating the initial apoptotic switch.

## 4. DISCUSSION

MASH ranks as the highest contributor to the global burden of chronic liver diseases and is becoming a public health challenge, mainly resulting from the obesity epidemic and limited therapeutic options [42,43,44]. In the present study, we assessed the preclinical effects of the 15-PGDH-inhibitor (SW033291) on MASH progression and its surrogates on metabolic and morphological disturbances in the HFD-mice MASH model. SW033291 administration reduced fat mass and increased lean mass, with a normalization of liver cell redox status that evaded insulin resistance. The normalization of metabolic patterns manifested morphologically as a decreased NAS score with regression of the liver fibrosis pattern and abrogation on the initial apoptotic switch. Metabolic effects appear to be mainly achieved in the mitochondrial citric acid cycle, probably by the downregulation of FOXO1, mTOR, and SIRT7 expressions.

Our results revealed a significant increase in TBW in the HFD and SW033291 groups compared with NMC at all timepoints. Nevertheless, SW033291-treated mice had a significant decrease in fat mass at week 24 when compared with their HFD counterparts. Additionally, the SW033291 group exhibited significantly higher lean mass and total body water compared with either NMC or HFD throughout the duration of this study. These findings indicate that SW033291 reduces body fat mass and improves lean body mass caused by high-fat diets. The above observation supports recent reports from preclinical studies that showed that SW033291-treated mice have lower fat mass and body weight [30,45].

The development of MASH signals a decline in cell redox manifested by a decrease in GSH [46,47,48], increased oxidized glutathione (GS:SG), and OA [46,49,50,51]. In our study, there was a decline in the cell redox status of the mice in the HFD group, as confirmed by significant increases in GSH, GS:SG, and OA levels compared with NMC at week 24. Interestingly, SW033291-treated mice had decreased GSH and GS:SG levels that returned to normal at week 24, but OA remained persistently elevated. Our data suggest that SW033291 improved, at least in part, the cell redox status that occurs in MASH progression. Blood glucose level, a surrogate of insulin resistance [52,53], was higher in the HFD group when compared with NMC at weeks 20 and 24. Although the blood glucose levels were initially elevated in the SW033291 group at week 16 of the study, blood glucose regressed at weeks 20 and 24, and they were not different from the NMC blood glucose level. These findings indicate that SW033291 may reduce insulin resistance, as others had found in a preclinical mouse model of type 2 diabetes mellitus [30,45]. This may be due to an increase in metabolic active lean mass, with a concomitant decrease in fat mass. Lipid β-oxidation in the mice was measured by the octanoate/butyrate ratio as a surrogate [54,55,56]. Our data show that the HFD group demonstrated a significantly higher octanoate/butyrate ratio in week 24 compared with NMC, depicting diminished lipid β-oxidation. However, treatment with SW033291 did not improve this state since there was no significant difference between the high-fat diet-fed mice and the SW033291-treated mice group (Figure 4b). These findings indicate that SW033291 treatment had no effect on mitochondrial lipid β-oxidation.

The metabolomics analysis expanded our observation, where SW033291 treatment significantly lowered the plasma concentrations to normal values of GS:SG, citric acid, and corticosterone (similar to the NMC group), which were significantly higher in the HFD mice. The heightened level of GS:SG in the HFD mice (Figure 3a,b) is in consonance with previous studies [46,49], suggesting that SW033291 potentially normalizes cell redox in MASH. Additionally, the HFD mice showed a significant increase in plasma citric acid concentration compared with NMC and SW033291-treated mice; high levels of citric acid have been reported in MASLD/MASH patients as well as in in vivo studies [57]. An increase in citric acid in these patients has been shown to correlate positively with oxidative stress, driving MASH progression [57]. Thus, normalizing citric acid through a more efficient citric acid cycle may be a mechanism through which SW033291 improves the cell redox status of the mice and exerts its anti-MASH potential. Glucocorticoids, including corticosterone, have been implicated as one of the major drivers in the pathogenesis and progression of MASLD/MASH, and their modulation is viewed as a putative strategy for the treatment of MASLD/MASH [58] Our data revealed SW033291’s ability to reduce corticosterone levels, pointing to its prospective potential in the treatment of MASH.

Previous studies have implicated proteins that play key roles in liver metabolism in the molecular pathogenesis of MASH [59,60,61,62]. *FOXO1* is a key transcriptional regulator of insulin actions. Insulin suppresses FOXO1 activity by activating the PI3K-AKT pro-survival signaling pathway. The activation of AKT phosphorylates FOXO1, promoting its nuclear extrusion, which leads to the inhibition of transcription [63,64]. FOXO1, when activated, is translocated into the nucleus and upregulates genes participating in glucose metabolism to increase gluconeogenesis [63,64]. In our work, the expression of FOXO1 in the HFD group was comparable to the NMC group throughout the study period. Interestingly, the SW033291-treated mice exhibited a significantly lower FOXO1 expression compared with NMC and HFD at weeks 16 and 20, as well as a lower expression than NMC at week 24. *mTOR*, a protein kinase that regulates energy balance, cellular growth, and metabolism [65], has been correlated with a significant improvement in insulin resistance and MASH [66,67]. In this study, significant changes in mTOR expression were observed at weeks 16 and 24. Mice treated with SW033291 had a significant reduction in mTOR expression as compared with the NMC and HFD groups. The lower expression of mTOR may indicate its role in reducing insulin resistance in the SW033291-treated group. Sirtuins are key regulators of liver lipid and carbohydrate metabolism, insulin signaling, and inflammatory responses [68,69]. We observed a significant decrease in *SIRT7* at week 24 in the SW033291-treated mice in comparison to the HFD and NMC groups (Figure 6c). SIRT7 is localized in the nucleus and is associated with the modulation of cell survival [59,69]. Studies using a genetic mouse model showed that liver-specific SIRT7 knockout mice were unsusceptible to high-fat diet-induced fatty liver, obesity, and glucose intolerance, with protection from fatty acid accumulation in the liver [59]. SW033291 may abrogate MASH progression by an aggregate metabolic effect targeting the citric acid cycle by regulating the expression levels of FOXO1, mTOR, and SIRT7.

MASH progression was quantitated by the NAS score, and our data revealed an increase in NAS score in the HFD-fed mice, which was partially rescinded in the SW033291-treated mice at 24 W, with regression of the liver fibrotic pattern and paucity of cell apoptotic activity. Our findings support the well-established observation that the hallmark of MASH is the accumulation of lipids in the liver (hepatic steatosis) [70]. Apoptosis is one of the most distinct features of MASH progression [52,71,72] since parenchymal apoptosis correlates with the degree of hepatic injury [52,71] and is the main cell activity that separates simple steatotic liver (MASLD) from MASH [52]. Nevertheless, one of the most striking findings was the resolution of the fibrotic pattern on the livers of mice treated with SW033291. This effect implies a normalization of hepatic stellate cell (HSC) collagen synthesis and deposition, with the degradation of the collagen exceeding that already deposited. Changes in HSC glycolytic metabolism through similar pathways previously described in parenchymal cells may explain, at least in part, HSC behavior [73,74]. In addition, concomitant resolution of the inflammatory component and cytokine environment could also play a role in the reversal of the fibrotic pattern [75,76,77] The present study did not aim to elucidate the effect of SW033291 on HSCs, and this question will be a matter of future research. The above findings indicate that SW0033291 alleviated inflammation and apoptosis, thus abrogating the initial apoptotic switch from MASH progression with a resolution of liver fibrosis.

During the last decades, the protective function of PGE2 in the liver has been described during ischemia–reperfusion injuries (IRI). Additionally, PGE2 has been recognized as a modulator of drug-induced liver injury and, thus, a putative therapeutic agent to reduce drug–liver toxicity [78]. As with IRI and drug-induced liver toxicity, oxidative stress is a hallmark of MASH [79,80]. Oxidative stress occurs when the production of reactive oxidative intermediates (ROIs) overwhelms the antioxidant capacity of the cell. In the liver, ROIs are mainly produced in mitochondria, as well as the endoplasmic reticulum of hepatocytes, by the activities of the cytochrome P450 enzymes. This process has also been described in macrophages and activated neutrophils [79,80,81]. The principal ROIs in the body include superoxide anions (O_2_−), hydroxyl radicals (OH·), and hydrogen peroxide (H_2_O_2_) [79,82]. ROIs are highly toxic in nature, but the body protects itself against their actions by detoxifying them via the antioxidant system. The major antioxidant system includes enzymes such as superoxide dismutase, glutathione peroxidase, reductase, and catalase, as well as non-enzymatic molecules, e.g., ascorbic acid, glutathione, retinol, and tocopherol and other proteins such as heme oxygenase-1 (HO-1) and redox proteins [79]. Together, these systems protect the body against the deleterious effects of ROIs [79]. Due to its pivotal role in metabolism, the liver is positioned to be particularly vulnerable to oxidative stress. Therefore, the liver is endowed with a unique ROI scavenging system that has nuclear factor E2-related factor 2 (Nrf2) as the central regulator [79,80]. Physiologically, (Nrf2) is in the cytoplasm, bound to the cytoskeletal-anchoring protein Kelch-like ECH-associated protein 1 [79,80]. However, during periods of heightened levels of ROIs, it is released and translocated to the nucleus, inducing the transcription of antioxidant genes under the control of antioxidant response elements (AREs) [79,80]. Additionally, Nrf2 plays a key role in regulating glutathione levels and acts as a positive modulator of the GSH/GSSG, as well as controlling the expression of enzymes that detoxify molecules in different organelles of the hepatocytes [79,80]. Upon reperfusion, there is increased hydrolysis of arachidonic acid (AA), the principal precursor of PGE2, via the catalytic action of phospholipase A2 (PLA2) and cyclooxygenase-2 (COX-2) [78]. PGE2 is produced in the liver by hepatocytes, and Kupffer cells and endothelial cells reduce inflammation and fibrosis [78]. PGE2 is degraded in vivo by 15-PGDH into an inactivated 15-keto-PGE2. Zhang et al. developed SW033291, an effective inhibitor of the degrading enzyme 15-PGDH [25]. Nevertheless, the exact mechanism for liver protection of PGE2 from oxidative stress remains to be determined.

Insulin resistance aggravates cell stress in MASH [83] by affecting inflammatory pathways from gradual mitochondrial dysfunction, leading to an increasing organ metabolic disturbance [84]. The generally accepted “two hit hypothesis” enunciates that hepatic steatosis (first hit) and insulin resistance (second hit) in conjunction modulated an increasing cell oxidative stress promoting the progression of MASH and collagen deposition (fibrosis) [17,18,85]. The action of this “second hit” leads to further hepatic lipotoxicity and aggravates the local inflammatory response [86]. The response is mediated by the release of danger-activated and pathogenic-activated molecular patterns that amplify the immune activation of liver resident macrophages (Kupffer cells), neutrophils, monocytes, dendritic cells, natural killer cells, and natural killer T cells [86]. The activated immune cells, in turn, further amplify their responses by producing cytokines and chemokines (e.g., TNF and IL-1β) and free radicals perpetuating the progression of MASH [86]. We had shown in a preclinical MASH model that SW033291 rescinded insulin resistance without any effect on β-lipid oxidation, restoring metabolic mitochondrial function and generating a paucity of apoptosis and the resolution of collagen deposition.

The branching of systematically studying the effects of SW033291on each cell and their interaction with local and systemic immunoregulatory cells will be the focus of future research. Nevertheless, previous work using a preclinical model of hepatic ischemic-reperfusion injury has shown that SW033291 extenuated inflammatory responses by lowering the expression of pro-inflammatory cytokines such as IL-1b, IL-6, and TNF-a from liver resident macrophages (Kupffer cells), nucleotide-binding oligomerization domain-like receptor protein 3 inflammatory vesicles (NLRP3), and chemokine 1 (CXCL1) in the serum and liver tissues of treated mice [29]. Additionally, it decreases macrophage and neutrophil infiltration in hepatic tissue, leading to the overall inhibition of inflammatory responses [29]. Former pro-inflammatory cytokines play key roles in the pathogenesis of liver fibrosis, and their inhibition by SW033291 may be one of the mechanisms through which it exhibits its anti-fibrotic effect [87].

The present findings must be perceived under the limitations of our study. The molecular mechanism(s) and signaling pathway(s) through which SW033291 exerts its anti-MASH effects were not directly elucidated, although its effects on the expression of metabolic proteins were determined. We did not compare SW033291 to other anti-MASH agents to evaluate its relative efficacy or safety profile. Although our results showed that SW033291 exercises its effects by positively modulating liver metabolic pathways with resolution of insulin resistance and apoptotic activity on parenchymal cells, we did not explicitly test to establish whether this was the mechanism that took place on HSCs for the regression of liver fibrosis. Furthermore, this was a 24 week duration study; thus, the long-term effects of SW033291 administration could not be ascertained. Additionally, we used only female mice for this study; therefore, we are unable to determine if sex differences could affect the effects of SW033291. Furthermore, although inhibition of 15-PGDH appears to positively modulate MASH-related metabolic and morphological changes in this study, its role in malignant disease (HCC) must be noted. Interestingly, studies on malignancies (liver, colon, and pancreas) have shown 15-PDGH inhibition or elevated PGE2 levels to drive tumor progression [88,89,90,91,92,93].

## 5. CONCLUSIONS

Based upon the results of the present study, the inhibition of 15-PDGH has the potential of abrogating MAS progression through a paucity of the initial apoptotic switch restoring physiological collagen’s microenvironment, with the resolution of insulin resistance status, oxidative stress, inflammation, and hepatic steatosis. The effects of SW033291 on HSCs and local immune-regulatory cells (Kuffer’s cell) remain to be determined. Future studies of SW033291 are warranted to determine its translational potential in MASH treatment.

## Figures and Tables

**Figure 1 cells-14-00987-f001:**
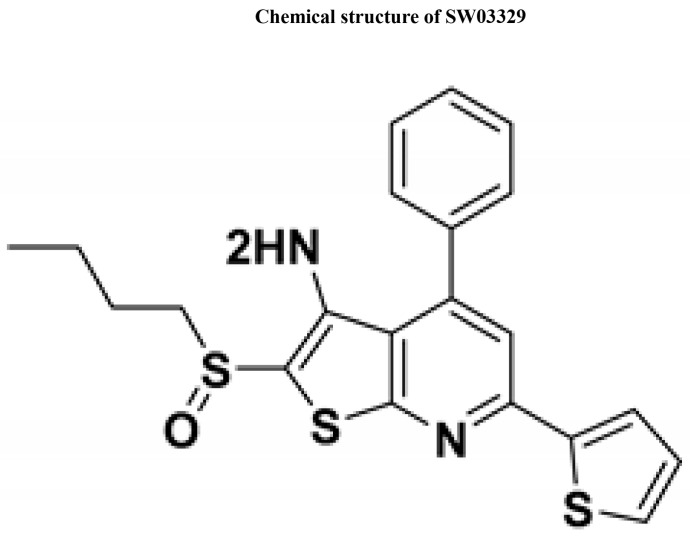
*Chemical structure of SW033291.* The chemical structure of SW033291 (inhibitor of 15-hydroxyprostaglandin dehydrogenase, adapted from reference [27]).

**Figure 2 cells-14-00987-f002:**
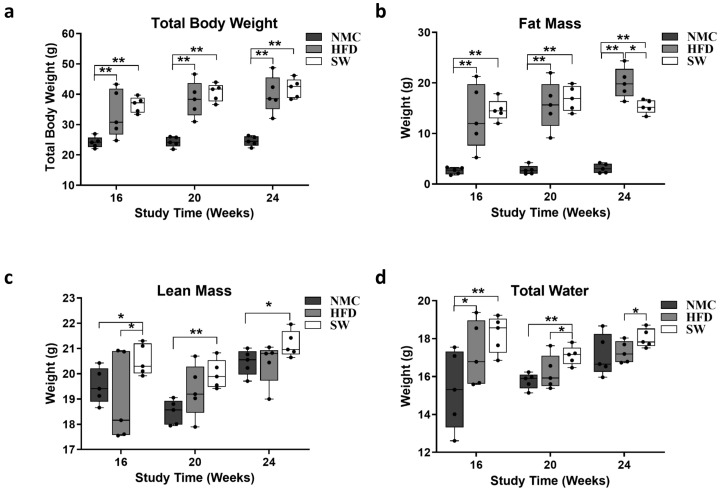
Mice total body composition. (**a**) Mice exposed to HFD with or without SW033291 treatment had increased total body weight compared with those in the NMC group at the timepoints in the experiment (** *p* < 0.01, by ANOVA and Tukey’s post hoc test/*t*-test, *n* =5). (**b**) The fat mass of the mice in the HFD and SW033291 groups increased simultaneously at 16 and 20 W when compared with their NMC group counterparts; however, at 24 W, there was a significant decrease in fat mass in the SW033291-treated mice when compared with the HFD group (* *p* < 0.05, ** *p* < 0.01, by ANOVA and Tukey’s post hoc test/*t*-test, *n* = 5). (**c**) The lean mass in the SW033291-treated mice increased when compared with the NMC-fed mice and to that of the HFD-fed mice at 16 W (* *p* < 0.05, ** *p* < 0.01). (**d**) Similarly, the total body water of the SW033291-treated mice increased significantly at the experimental timepoints when compared with the HFD-fed mice and the NMC-fed mice at 16 and 20 W (* *p* < 0.05, ** *p* < 0.01).

**Figure 3 cells-14-00987-f003:**
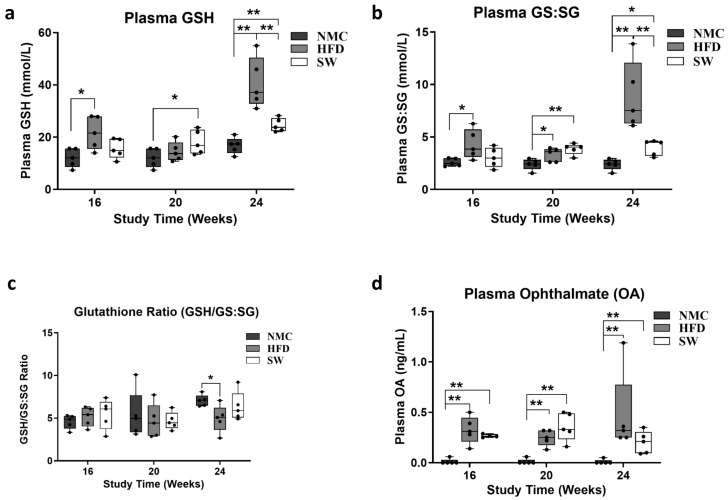
Cell redox assessment in plasma from mice with MASH. There were significant differences in the glutathione sp. (GSH, GS:SG) and OA among groups and across experimental timepoints. (**a**) Mice in the HFD group had a significantly higher concentration of GSH when compared with the NMC at 16 and 24 W. The GSH level in the SW033291-treated mice decreased significantly by week 24 (* *p* < 0.05, ** *p* < 0.01, by ANOVA and Tukey’s post hoc test/*t*-test, *n* = 5). (**b**) GS:GS increased in the HFD group compared with the NMC group at all timepoints in the experiment, an effect that was rescinded by week 24 in the SW033291 mice (* *p* < 0.05, ** *p* < 0.01). (**c**) There was no significant difference in the glutathione ratio (GSH/GS:SG) among groups and experimental timepoints, except at week 24, where there was a significant decrease in the HFD group in comparison to the NMC group (*stands for *p* < 0.05). (**d**) The OA levels were significantly higher in both the HFD and SW033291 groups when compared with NMC at all timepoints (** *p* < 0.01).

**Figure 4 cells-14-00987-f004:**
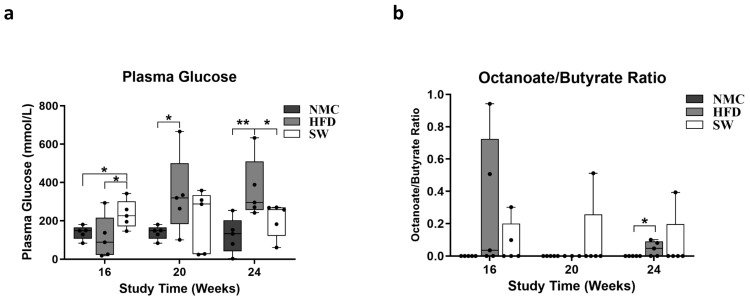
Insulin-resistant status and lipid β-oxidation assessment in MASH. (**a**) Glucose concentrations in the HFD group were higher than in the NMC group at all timepoints in the experiment. In contrast, the was an initial significant increase in the blood glucose level of the SW033291-treated mice at week 16, but this was significantly lowered down to a normal value at 24 W (* *p* < 0.05, ** *p* < 0.01, by ANOVA and Tukey’s post hoc test/*t*-test, *n* = 5). (**b**) The octanoate/butyrate concentration ratios in plasma (surrogate for β-oxidation) were not significantly different among groups at all experimental time points; At week 24, there was a significant increase in the ration form the HFD mice when compared with the NMC-diet mice, an effect that was not ameliorated by SW033291 treatment (* *p* < 0.05).

**Figure 5 cells-14-00987-f005:**
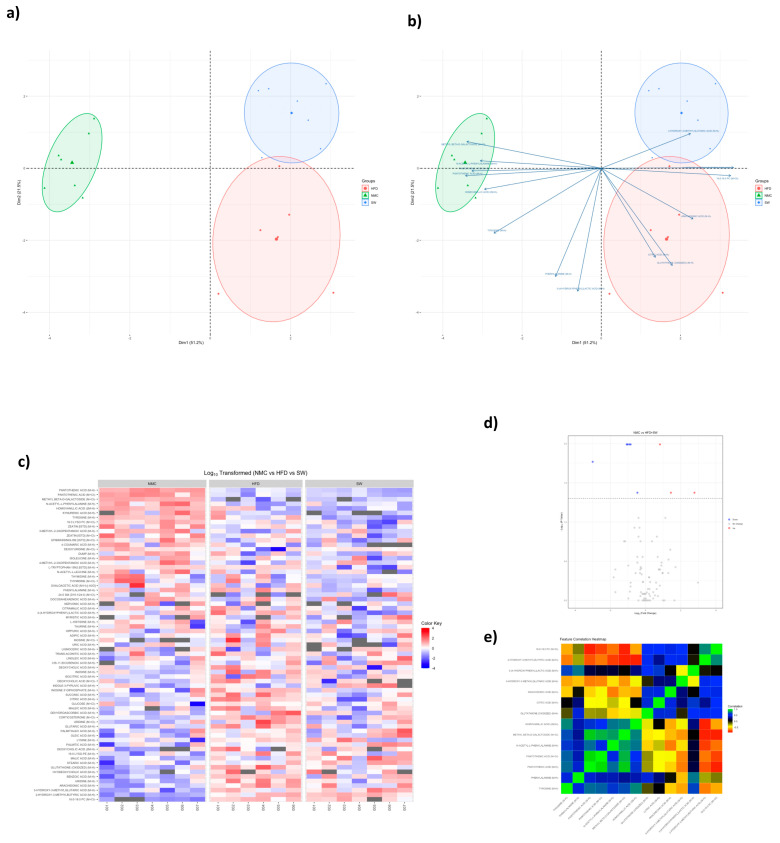
PCA plots of plasma non-targeted metabolomics in MASH mice. Plasma non-targeted metabolomics data in the NMC, HFD, and SW033291−treated mouse groups, showing variations in metabolic profiles (**a**) Principal component analysis (PCA) plot displaying the differences in metabolites among groups. (**b**) Biplot showing principal component scores (points) and variable loadings (arrows). (**c**) Heat map showing discrimination in metabolites in mice serum (NMC vs. HFD vs. SW033291). (**d**) Volcano plot for SW033291-treated mice compared with NMC. (**e**) Correlation heat map displaying the relationship between each metabolite across the experimental groups. (*n* = 5, data analyzed with Kruskall-Wallis and Benjamin-Hochberg false discovery rate (FDR) correction).

**Figure 6 cells-14-00987-f006:**
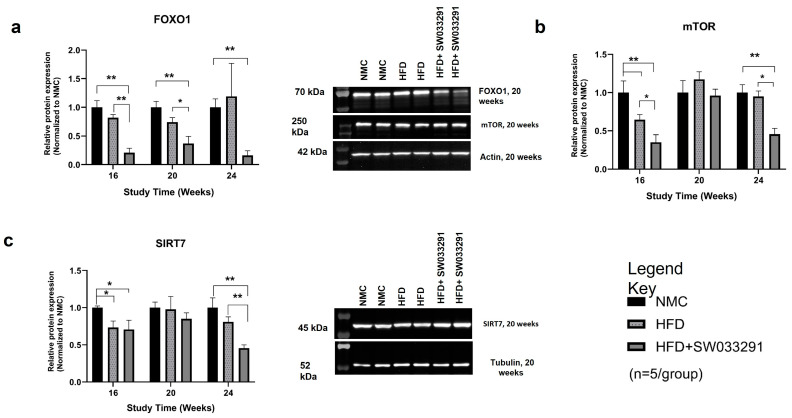
Effects of SW033291 on protein expression levels in mice. (**a**) FOXO1 expression was significantly decreased in the HFD + SW033291 group compared with both the NMC and HFD groups at 16 and 20 weeks and compared with the NMC at 24 weeks (* *p* < 0.05, ** *p* < 0.01, by ANOVA with Tukey’s post hoc test or unpaired *t*-test, *n* = 5/group). (**b**) Expression of mTOR was significantly lower in the HFD + SW033291 group compared with the NMC and HFD groups at 16 and 24 weeks (* *p* < 0.05, ** stands for *p* < 0.01). (**c**) SIRT7 expression was significantly reduced in the HFD + SW033291 group compared with both the NMC and HFD at 24 weeks and compared with NMC at 16 weeks (* *p* < 0.05, ** *p* < 0.01). Black bars represent NMC; gray patterned bars represent HFD; solid dark gray bars represent HFD+ SW033291. Representative Western blots for each protein at week 20 are shown beside each graph; FOXO1 and mTOR were normalized to actin; SIRT7 was normalized to tubulin.

**Figure 7 cells-14-00987-f007:**
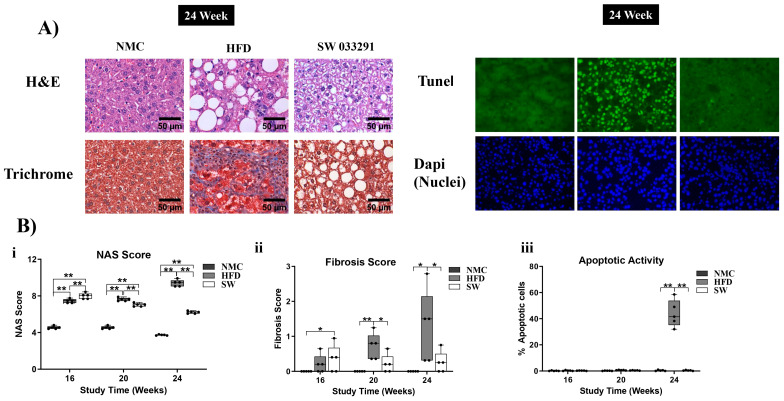
Histological assessment of livers with MASH at 24 W. (**A**) There was pronounced steatosis, fibrosis, and apoptosis in the HFD mice compared with both the NMC and SW033291-treated mice (representative H&E, Trichrome, and TUNEL staining images). (**Bi**) The NAS score was significantly higher in the HFD and SW033291 groups in comparison to the NMC group at all experimental timepoints except for 24 W, when there was a significant reduction in the NAS score in the SW033291-treated group (** *p* < 0.01, by ANOVA and Tukey’s post hoc test/*t*-test, *n* = 5). (**Bii**) There was increased fibrosis at 20 and 24 W in the HFD-fed mice in comparison to both the NMC and the SW033291-treated mice. (**Biii**) There was no significant difference in the apoptotic activity among groups at 16 and 20 W. Albeit, we observed a significant increase in apoptosis in the HFD group when compared with both the NMC-fed mice and the SW0033291-treated mice at 24 W. * stands for *p* < 0.05, while ** stands for *p* < 0.01.

**Table 1 cells-14-00987-t001:** (**a**) Plasma concentrations of metabolites (class of amino acids; mmol/L) at week 24. (**b**) Plasma concentration of metabolites (class of carbohydrates; mmol/L) at week 24. (**c**) Plasma concentration of metabolites (class of lipids, mmol/L) at week 24. (**d**) Comprehensive metabolomics; µmol/L. * denotes *p*-value significant at <0.05.

**(a)**														
		**Experimental Group**												
**METABOLITE NAME**		**Class**			**NMC**			**HFD**		**NMC vs. HFD**	**SW033291**			**HFD vs. SW033291**
					24 Week			24 Week		*p*-value *	24 Week			*p*-value *
GLUTATHIONE (OXIDIZED)		Amino acid			19.58 ± 12.0			58.31 ± 28.2		NS	25.95 ± 7.9			<0.05
HIPPURIC ACID		Amino acid			22,131.74 ± 19.9			22,456.87 ± 2210.2		NS	21,955.22 ± 2117.0			NS
ISOLEUCINE		Amino acid			212.67 ± 45.4			186.67 ± 24.3		NS	165.58 ± 50.2			NS
L-HISTIDINE		Amino acid			8.28 ± 3.3			7.55 ± 3.2		NS	8.21 ± 4.1			NS
L-TRYPTOPHAN		Amino acid			526.58 ± 58.2			521.39 ± 52.2		NS	247.50 ± 215.5			<0.05
LYSINE		Amino acid			8.44 ± 2.2			10.98 ± 0.9		NS	10.83 ± 3.8			NS
N-ACETYL-L-LEUCINE		Amino acid			130.31 ± 44.5			105.02 ± 29.9		NS	97.87 ± 28.6			NS
N-ACETYL-L-PHENYLALANINE		Amino acid			61.05 ± 11.8			38.19 ± 6.4		<0.05	42.01 ± 6.0			NS
PHENYLALANINE		Amino acid			341.33 ± 50.1			364.37 ± 46.3		NS	271.41 ± 25.8			<0.05
TYROSINE		Amino acid			223.15 ± 20.1			184.14 ± 63.3		NS	102.57 ± 25.4			<0.05
**(b)**														
	**Experimental Group**													
**METABOLITE NAME**	**Class**		**NMC**			**HFD**	**NMC vs. HFD**	**SW033291**				**HFD vs. SW033291**		
			24 Week			24 Week	*p*-value *	24 Week				*p*-value *		
BENZOIC ACID	Carbohydrate		52.51 ± 18.5			96.71 ± 6.4	NS	79.76 ± 31.4				NS		
CITRAMALIC ACID	Carbohydrate		1700.75 ± 578.1			1864.37 ± 856.9	NS	1347.37 ± 566.5				NS		
CITRIC ACID	Carbohydrate		2708.11 ± 201.4			3746.76 ± 460.4	<0.05	2775.55 ± 669.8				<0.05		
GLUCOSE	Carbohydrate		282.89 ± 282.0			345.56 ± 150.7	NS	368.92 ± 308.1				NS		
GLUTARIC ACID	Carbohydrate		23.68 ± 7.3			47.88 ± 19.2	NS	27.78 ± 12.5				<0.05		
HOMOVANILLIC ACID	carbohydrate		13.69 ± 1.0			10.05 ± 3.1	<0.05	7.64 ± 1.5				<0.05		
INDOLE-3-PYRUVIC ACID	carbohydrate		5357 ± 3622			4.77 ± 2.6	NS	15.25 ± 9.7				NS		
ISOCITRIC ACID	carbohydrate		3562.14 ± 250.8			4110.98 ± 995.2	NS	3773.45 ± 559.7				NS		
MALEIC ACID	carbohydrate		42.93 ± 15.0			73.77 ± 24.5	<0.05	48.92 ± 19.2				NS		
MALIC ACID	carbohydrate		46.12 ± 11.0			57.99 ± 16.3	NS	92.63 ± 36.6				NS		
OXALOACETIC ACID	carbohydrate		412.77 ± 48.4			388.33 ± 39.3	NS	391.72 ± 21.9				NS		
SUCCINIC ACID	carbohydrate		20.83 ± 153.9			551.80 ± 356.4	NS	230.69 ± 179.6				<0.05		
TRANS-ACONITIC ACID	carbohydrate		234.20 ± 210.3			313.10 ± 221.0	NS	231.61 ± 150.4				NS		
**(c)**														
	**Experimental Group**													
	**Class**		**NMC**			**HFD**	**NMC vs. HFD**	**SW033291**				**HFD vs. SW033291**		
**METABOLITE NAME**			24 Week			24 Week	*p*-value *	24 Week				*p*-value *		
ADIPIC ACID	Lipid		9.02 ± 2.1			14.02 ± 5.7	NS	9.59 ± 7.5				NS		
CAPRYLIC ACID	Lipid		56.22 ± 20.2			187.55 ± 89.1	NS	135.93 ± 106.6				NS		
CIS-11-EICOSENOIC ACID	Lipid		6.65 ± 1.9			7.08 ± 2.2	NS	8.33 ± 3.5				NS		
CORTICOSTERONE	Lipid		51.34 ± 5.7			85.03 ± 11.3	<0.05	55.76 ± 21.3				<0.05		
DOCOSAHEXAENOIC ACID	Lipid		140.46 ± 91.9			125.75 ± 111.5	NS	111.24 ± 130.0				NS		
EPIBRASSINOLIDE	Lipid		430.64 ± 36.3			392.40 ± 39.4	<0.05	377.29 ± 50.7				NS		
LIGNOCERIC ACID	Lipid		2.02 ± 0.7			1.95 ± 0.8	NS	2.40 ± 0.2				NS		
LINOLEIC ACID	Lipid		102.82 ± 66.7			114.24 ± 34.9	NS	129.82 ± 114.1				NS		
MYRISTIC ACID	Lipid		4.04 ± 1.5			3.55 ± 1.0	NS	4.19 ± 2.0				NS		
NERVONIC ACID	Lipid		1.96 ± 0.5			1.8 ± 0.3	NS	1.77 ± 0.2				NS		
OLEIC ACID	Lipid		177.67 ± 100.0			236.90 ± 51.5	NS	296.92 ± 178.9				NS		
PALMITIC ACID	Lipid		56.99 ± 32.5			63.56 ± 8.9	NS	97.62 ± 40.2				NS		
SEBACIC ACID	Lipid		20.57 ± 16.1			27.06 ± 7.9	NS	35.80 ± 26.9				NS		
STEARIC ACID	Lipid		57.71 ± 20.6			75.40 ± 15.2	NS	93.62 ± 23.5				NS		
**(d)**														
**Metabolite Name**				**NMC**			**HFD**		**SW033291**			***p*-value**	**Benjamini–Hochberg corrected *p*-value**	
16:0 LYSO PC (M+Cl)-				999,071 ± 67,793			871,748 ± 65,482		787,031 ± 163,053			0.0137	NS	
16:0–18:0 PC (M+Cl)-				47,443 ± 5741			140,776 ± 25,533		132,632 ± 29,631			0.0049	0.0441	
18:0 LYSO-PE (M-H)-				141,905 ± 76,160			306,758 ± 169,087		251,645 ± 148,057			NS	NS	
20:0 LYSO PC (M+Cl)-				9055 ± 4134			12,829 ± 9749		8540 ± NA			NS	NS	
24:0 SM (D18:1/24:0) (M+Cl)-				72,815 ± 16,580			42,977 ± 9509		75,725 ± 30,083			0.0416	NS	
2-HYDROXY-3-METHYLBUTYRIC ACID (M-H)-				136,900 ± 14,513			238,877 ± 40,419		253,001 ± 24,659			0.0014	0.0336	
2-HYDROXYBUTYRIC ACID (M-H)-				NaN ± NA			NaN ± NA		48,141 ± NA			NP	NP	
3-(4-HYDROXYPHENYL)LACTIC ACID (M-H)-				233,017 ± 53,067			306,087 ± 80,924		159,901 ± 56,999			0.0066	0.0464	
3′-CMP (M-H)-				NaN ± NA			3695 ± 3003		6276 ± NA			NS	NS	
3-HYDROXY-3-METHYLGLUTARIC ACID (M-H)-				31,353 ± 25,717			102,719 ± 56,648		153,170 ± 114,368			0.0069	0.0464	
3-METHYL-2-OXOPENTANOIC ACID (M-H)-				1,062,442 ± 174,392			809,583 ± 202,873		859,039 ± 160,885			NS	NS	
3-UREIDOPROPIONIC ACID (M-H)-				13,633 ± 4566			27,704 ± 25,280		23,812 ± 6769			NS	NS	
4-COUMARIC ACID (M-H)-				31,824 ± 14,756			15,533 ± 8403		14,592 ± 10,303			NS	NS	
4-HYDROXYBENZOIC ACID (M-H)-				9132 ± 1185			5225 ± NA		NaN ± NA			NS	NS	
4-METHYL-2-OXOPENTANOIC ACID (M-H)-				651,191 ± 187,117			468,621 ± 142,524		561,698 ± 137,764			NS	NS	
ADIPIC ACID (M-H)-				9023 ± 2135			14,024 ± 5700		9589 ± 7496			NS	NS	
ARACHIDIC ACID (M-H)-				17,808 ± 7593			16,941 ± 10,347		10,695 ± 5785			NS	NS	
ARACHIDONIC ACID (M-H)-				56,224 ± 20,232			187,550 ± 89,136		135,930 ± 106,642			0.0047	0.0441	
BEHENIC ACID (M-H)-				17,921 ± 2014			14,981 ± 3108		13,695 ± 3147			NS	NS	
BENZOIC ACID (M-H)-				52,510 ± 18,477			96,710 ± 6365		79,761 ± 31,390			0.0213	NS	
CAPRYLIC ACID (M-H)-				8235 ± NA			14,695 ± 5273		91,301 ± 71,223			NS	NS	
CIS-11-EICOSENOIC ACID (M-H)-				6653 ± 1896			7079 ± 2194		8326 ± 3508			NS	NS	
CITRAMALIC ACID (M-H)-				1,700,753 ± 578074			1,864,369 ± 856,907		1,347,370 ± 566,523			NS	NS	
CITRIC ACID (M-H)-				2,708,109 ± 201,367			3,746,763 ± 460,431		2,775,550 ± 669,761			0.0022	0.036	
CORTICOSTERONE (M+Cl)-				51,373 ± 5709			85,029 ± 11,330		55,762 ± 21,315			0.0119	NS	
DEHYDROASCORBIC ACID (M-H)-				135,278 ± 73,273			348,721 ± 319,377		203,330 ± 91,806			NS	NS	
DEOXYCHOLIC ACID (2M-H)-				2967 ± 2017			7326 ± 6067		4896 ± 1959			NS	NS	
DEOXYCHOLIC ACID (M+Cl)-				34,177 ± 12,878			55,698 ± 25,533		40,464 ± 18,047			NS	NS	
DEOXYCHOLIC ACID (M-H)-				113,724 ± 46,601			199,481 ± 105,148		133,817 ± 69,673			NS	NS	
DEOXYURIDINE (M+Cl)-				154,902 ± 23,677			103,335 ± 48,760		131,551 ± 13,136			NS	NS	
DEOXYURIDINE (M-H)-				32,078 ± NA			29,972 ± 7838		67,584 ± NA			NS	NS	
D-GLUCOSAMINE 6-SULFATE (M-H)-[-H2O]				18,374 ± 8325			17,577 ± 1312		18,702 ± 24,299			NS	NS	
DOCOSAHEXAENOIC ACID (M-H)-				140,460 ± 91,921			125,748 ± 111,514		111,240 ± 130,038			NS	NS	
DUMP (M-H)-				47,184 ± 11,690			37,025 ± 15,323		36,035 ± 11,244			NS	NS	
EPIBRASSINOLIDE [ISTD] (M+Cl)-				430,639 ± 36,260			392,399 ± 39,397		377,287 ± 50,648			NS	NS	
ERUCIC ACID (M-H)-				6393 ± 3349			7132 ± 3663		4177 ± 2520			NS	NS	
FLAVIN ADENINE DINUCLEOTIDE (M-H)-				5672 ± 2621			12,029 ± NA		6807 ± NA			NS	NS	
GLUCOSE (M+Cl)-				282,885 ± 282,005			345,560 ± 150,718		368,916 ± 308,115			NS	NS	
GLUTAMINE (M-H)-				22,951 ± 9624			20,569 ± 9510		25,922 ± 17,703			NS	NS	
GLUTARIC ACID (M-H)-				23,683 ± 7280			47,883 ± 19,225		27,778 ± 12,511			0.0414	NS	
GLUTATHIONE (OXIDIZED) (M-H)-				19,578 ± 12,000			58,391 ± 28,237		25,952 ± 7853			0.0053	0.0441	
HEPTADECANOIC ACID (M-H)-				3545 ± 722			4146 ± 1220		3343 ± 972			NS	NS	
HIPPURIC ACID (M-H)-				22,131,742 ± 1,988,863			22,456,867 ± 2,210,184		21,955,215 ± 2,117,015			NS	NS	
HOMOVANILLIC ACID (2M-H)-				13,689 ± 1028			10,045 ± 3139		7643 ± 1505			0.0028	0.0336	
HYODEOXYCHOLIC ACID (M-H)-				74,010 ± 56,633			864,680 ± 1,358,876		748,160 ± 955,799			NS	NS	
INDOLE-3-PYRUVIC ACID (M-H)-				5357 ± 3622			4765 ± 2582		15,253 ± 9718			NS	NS	
INOSINE (M+Cl)-				5817 ± 3002			13,905 ± 17,966		4782 ± 2270			NS	NS	
INOSINE (M-H)-				26,120 ± 14,532			65,914 ± 52,076		29,321 ± 17,353			NS	NS	
INOSINE 5′-DIPHOSPHATE (M-H)-				74,867 ± 10,956			83,878 ± 3826		75,515 ± 4497			0.0293	NS	
ISOCITRIC ACID (M-H)-				3,562,142 ± 250,803			4,110,980 ± 995,231		3,773,454 ± 559,709			NS	NS	
ISOLEUCINE (M-H)-				212,672 ± 45,422			186,668 ± 24,327		165,577 ± 50,201			NS	NS	
KYNURENIC ACID (M-H)-				15,956 ± 5823			9186 ± 4342		7267 ± 1166			NS	NS	
L-HISTIDINE (M-H)-				8280 ± 3336			7551 ± 3216		8214 ± 4073			NS	NS	
LIGNOCERIC ACID (M-H)-				2016 ± 669			1951 ± 750		2400 ± 228			NS	NS	
LINOLEIC ACID (M-H)-				102,817 ± 66,691			114,235 ± 34,869		129,823 ± 114,118			NS	NS	
L-TRYPTOPHAN-15N2 [ISTD] (M-H)-				526,582 ± 58,182			521,388 ± 52,165		247,497 ± 215,531			0.0431	NS	
LYSINE (M-H)-				8474 ± 2194			10,975 ± 922		10,832 ± 3802			NS	NS	
MALEIC ACID (M-H)-				42,927 ± 15,026			73,767 ± 24,525		48,916 ± 19,167			0.0492	NS	
MALIC ACID (M-H)-				46,118 ± 11,016			57,993 ± 16,328		92,673 ± 36,594			0.0213	NS	
METHYL BETA-D-GALACTOSIDE (M+Cl)-				139,948 ± 15,875			15,084 ± 9067		18,351 ± 5849			0.0027	0.0336	
METHYLMALONIC ACID (M-H)-				18,467 ± 6095			21,842 ± 1848		16,695 ± 6137			NS	NS	
MYRISTIC ACID (M-H)-				4038 ± 1518			3553 ± 995		4185 ± 2014			NS	NS	
MYRISTOLEIC ACID (M-H)-				8435 ± 4072			10,032 ± 2060		5925 ± 5062			NS	NS	
N-ACETYLGLYCINE (M-H)-				19,086 ± 16,253			6539 ± NA		NaN ± NA			NS	NS	
N-ACETYL-L-ALANINE (M-H)-				2428 ± NA			20,516 ± 10,736		61,836 ± NA			NS	NS	
N-ACETYL-L-LEUCINE (M-H)-				130,301 ± 44,499			105,017 ± 29,922		97,866 ± 28,641			NS	NS	
N-ACETYL-L-PHENYLALANINE (M-H)-				61,046 ± 11,829			38,194 ± 6372		42,010 ± 6002			0.0029	0.0336	
NERVONIC ACID (M-H)-				1959 ± 482			1798 ± 334		1766 ± 165			NS	NS	
OLEIC ACID (M-H)-				177,669 ± 100,039			236,903 ± 51,510		296,922 ± 178,847			NS	NS	
OXALOACETIC ACID (M-H)-[-H2O]				412,772 ± 48,365			388,334 ± 39,291		391,721 ± 21,886			NS	NS	
PALMITIC ACID (M-H)-				56,987 ± 32,515			63,557 ± 8874		97,618 ± 40,172			NS	NS	
PALMITOLEIC ACID (M-H)-				20,565 ± 16,069			27,060 ± 7841		35,802 ± 26,925			NS	NS	
PANTOTHENIC ACID (M+Cl)-				33,766 ± 4352			19,796 ± 3323		18,831 ± 1855			0.0013	0.0336	
PANTOTHENIC ACID (M-H)-				319,857 ± 24,878			175,161 ± 26,571		158,159 ± 19,156			0.0011	0.0336	
PHENYLALANINE (M-H)-				341,328 ± 50,115			364,373 ± 46,304		271,410 ± 25,772			0.0071	0.0464	
PHTHALIC ACID (M-H)-				22,434 ± NA			11,568 ± 2642		12,377 ± 1856			NS	NS	
PIMELIC ACID (M-H)-				7642 ± 1255			8244 ± 360		11,072 ± NA			NS	NS	
SEBACIC ACID (M-H)-				28,480 ± 8595			NaN ± NA		7867 ± NA			NS	NS	
STEARIC ACID (M-H)-				57,709 ± 20,613			75,404 ± 15,206		93,624 ± 23,480			NS	NS	
SUCCINIC ACID (M-H)-				209,834 ± 153,853			551,798 ± 356,372		230,686 ± 179,562			NS	NS	
TAURINE (M-H)-				19,104 ± 6139			20,171 ± 6428		17,516 ± 3764			NS	NS	
THYMIDINE (M+Cl)-				97,088 ± 19,468			90,952 ± 15,171		84,750 ± 6181			NS	NS	
THYMIDINE (M-H)-				49,977 ± 10,993			45,691 ± 9062		42,271 ± 3163			NS	NS	
THYMINE-D4(METHYL-D3,6-D1) [ISTD] (M-H)-				263,379 ± 10,027			245,320 ± 22,888		246,512 ± 34,562			NS	NS	
TRANS-ACONITIC ACID (M-H)-				234,201 ± 210,269			313,096 ± 221,031		231,614 ± 150,408			NS	NS	
TYROSINE (M-H)-				223,150 ± 20,045			184,140 ± 63,255		102,568 ± 25,374			0.0018	0.0336	
URIC ACID (M-H)-				226,627 ± 245,663			597,501 ± 781,901		401,973 ± 423,309			NS	NS	
URIDINE (M+Cl)-				130,771 ± 73,528			303,107 ± 229,645		172,537 ± 93,266			NS	NS	
URIDINE (M-H)-				41,396 ± 14,784			95,222 ± 33,938		63,181 ± 18,945			0.0153	NS	
XANTHINE (M-H)-				32,567 ± NA			63,862 ± 58,166		53,646 ± 79,311			NS	NS	
XANTHOSINE (M-H)-				15,451 ± NA			23,618 ± 13,508		NaN ± NA			NS	NS	
ZEATIN [ISTD] (M+Cl)-				95,865 ± 13,149			87,654 ± 6336		79,413 ± 10,396			NS	NS	
ZEATIN [ISTD] (M-H)-				71,790 ± 11,292			62,437 ± 5294		56,018 ± 8993			0.045	NS	

Table 1d analysis carried out with Kruskal-Wallis test with Benjamini-Hochberg false discovery rate (FDR) correction. Highlighted metabolites (in yellow color), significant both with Kruskal-Wallis test and Benjamini-Hochberg Correction.

## Data Availability

The original contributions presented in this study are included in the article/Appendix A. Further inquiries can be directed to the corresponding author(s).

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
