# Peer review of "INHIBITION OF THE PROSTAGLANDIN-DEGRADING ENZYME 15-PGDH AMELIORATES MASH-ASSOCIATED APOPTOSIS AND FIBROSIS IN MICE"

_cells, 2025, doi:10.3390/cells14130987_

Round 1

Reviewer 1 Report

Comments and Suggestions for Authors

Comments to authors

  • Methodology section
  1. In the methodology section, the authors state that seven mice were used; however, only five are presented in the figure legends and results. Could the authors clarify whether any mice were excluded or if there were any deaths during the study?

  1. The authors exclusively used female mice for this study. However, the inclusion of male mice may be preferable. Female mice experience estrous cycle variations, leading to hormonal fluctuations that can introduce variability in experimental outcomes. These fluctuations may influence metabolism, immune response, and drug efficacy, potentially affecting the reproducibility and consistency of results.

  1. The authors recommended including a normal mice control group treated with SW033291 as a control in addition to HFD with SW033291 to test its effects on healthy donors.

  1. Whether a one-way or two-way ANOVA test was used in the study analysis is unclear. Could the authors clarify which ANOVA test was applied in the analysis?

  • Results section
  1. Since lipid accumulation is a key characteristic of MASH, the authors should have presented a comprehensive lipid profile, including total cholesterol, LDL, HDL, and triglyceride levels in serum or liver tissue. Additionally, staining liver tissue with Oil Red O, rather than H&E staining, would provide a more specific assessment of lipid accumulation, offering a clearer evaluation of the metabolic alterations associated with the disease model.

  1. The authors assessed serum glucose levels to evaluate the role of SW033291 in improving insulin resistance. However, to provide a more comprehensive assessment of insulin sensitivity and metabolic function, it would be necessary to include insulin levels, C-peptide, HOMA-IR calculations, and a glucose tolerance test (GTT). These additional parameters would offer a more robust evaluation of glucose metabolism and the therapeutic impact of the intervention on SW033291 resistance.

  1. Masson's Trichrome stain is a representative method for visualizing liver fibrosis; however, for a more precise and quantitative assessment of liver fibrosis, it would be beneficial to evaluate specific fibrotic markers such as α-SMA, collagen I, and desmin using techniques like Western blotting or RT-PCR. These molecular approaches provide more detailed insights into the degree of fibrogenesis and the underlying mechanisms involved.

  1. The current study did not assess the potential immunomodulatory effect of 15-PGDH inhibition via SW033291. Given the well-documented role of prostaglandins-particularly PGE2-in regulating T cell activation, NK cell cytotoxicity, and macrophage polarization, it would be valuable for the authors to explore or at least discuss the impact of SW033291 on immune cells involved in liver inflammation and fibrosis.

  • Discussion section
  1. The authors recommended discussing how their preclinical findings might translate to human trials or clinical endpoints.

Author Response

 Thank you so much for your kind and valuable review of our manuscript.  Responses to comments are contained in the attached document.

Reviewer 2 Report

Comments and Suggestions for Authors

The manuscript "Inhibition of the Prostaglandin-Degrading Enzyme 15-PGDH Ameliorates MASH-Associated Apoptosis and Fibrosis in Mice" investigates the effects of SW033291, a 15-PGDH inhibitor, on MASH (metabolic dysfunction-associated steatohepatitis) progression in a mouse model.

You should explain why inhibiting this enzyme is a promising therapeutic approach and further discuss oxidative stress, insulin resistance, and inflammation in greater detail.

Explain why SW033291 was administered at 5 mg/kg IP twice daily and how these durations were chosen based on disease progression models.

Discuss why SW033291 reduced apoptosis but not lipid β-oxidation and whether SW033291 affects non-hepatic prostaglandin pathways.

Discuss consistency or discrepancies with other studies on 15-PGDH inhibition.

Suggest how these findings could lead to human clinical trials.

Author Response

(The authors gave the same response as above.)

Reviewer 3 Report

Comments and Suggestions for Authors

This study of your mouse model for MAFLD is  important. You conclude that SW033291 effects on MASH is promising but its effect on regression of hepatic fibrosis remains to be proven. Are your scientists in West Virginia and Cleveland Ohio collaborating with other centers in translational studies evaluating SW033291 in human MASH patients?

 Rodent models for steatohepatitis developed in West Virginia have been utilized by many centers. This study is well done This paper will be of interest to a wide spectrum of readers including, caregivers, basic scientists , veterinarians, liver centers, endocrinologists and pharmaceutical companies. The references are important and timely (2024) Figures1-7, Tables 1a,b,c and supplemantal material are well done and helpful. No need for changes.

There is no evidence of any unethical issues including plagiarism, inappropriate self-citatios or conflicts of interest.

Author Response

(The authors gave the same response as above.)

Reviewer 4 Report

Comments and Suggestions for Authors

Title:

Consider shortening and clarifying the title to emphasize the key findings while maintaining specificity. For example:

“SW033291 Modulates Hepatic Metabolism and Morphology in a Mouse Model of MASH: A Preclinical Evaluation”

Abstract:

• Clarify the hypothesis: Explicitly state whether the study hypothesizes that SW033291 improves MASH pathology via 15-PGDH inhibition.

• Include methodological details: Mention the sample size and key experimental groups to provide context on study robustness.

• Refine conclusions: Soften claims such as “ameliorates metabolic and morphological hepatic abnormalities” by acknowledging limitations, e.g., “demonstrates potential for modulating metabolic and morphological changes in MASH, warranting further investigation.”

Introduction:

• Streamline background information: Reduce excessive epidemiological data and focus on what is unknown about 15-PGDH inhibition in MASH.

• Strengthen research gap: Clearly articulate the limitations of current MASH treatments and why SW033291 presents a novel therapeutic angle.

• Highlight novelty: Differentiate this study from previous work on SW033291, emphasizing new insights into metabolic pathways and disease progression.

Materials & Methods:

• Control for confounders: Explicitly state how factors like diet composition and baseline metabolic differences were managed.

• Justify sample size: Provide a power analysis or rationale for the number of mice used.

• Ensure reproducibility: Mention whether histological assessments and metabolic measurements were performed in a blinded or randomized manner.

• Detail statistical methods: Clarify how multiple comparisons were addressed (e.g., Bonferroni correction) and whether normality assumptions were tested.

Results:

• Improve data presentation: Simplify metabolomics data presentation for clarity and accessibility.

• Report effect sizes: Include Cohen’s d or confidence intervals alongside p-values to enhance biological interpretation.

• Address variability: Discuss individual variability (e.g., range, outliers) rather than relying solely on mean ± SD.

Discussion:

• Avoid overinterpretation: Reframe definitive claims (e.g., “SW033291 restores metabolic function”) with more measured language (e.g., “SW033291 modulates metabolic pathways, suggesting a potential role in MASH treatment.”).

• Compare to existing therapies: Benchmark SW033291’s effects against standard MASH treatments to provide clinical context.

• Acknowledge side effects: Discuss potential toxicity, long-term effects, and metabolic trade-offs.

• Address species differences: Highlight limitations in translating findings from mouse models to human MASH pathology.

Conclusion:

• Soften definitive claims: Instead of stating that SW033291 “ameliorates MASH”, consider “shows promise in modulating MASH-related metabolic and morphological changes.”

• Provide future directions: Suggest next steps, such as testing in alternative models, exploring long-term effects, or conducting mechanistic studies.

Tables & Figures:

• Enhance clarity: Improve figure labels, axis annotations, and font sizes for readability.

• Increase resolution: Ensure figures are high-resolution for clarity.

• Include statistical annotations: Clearly indicate statistical significance in tables to aid interpretation.

Comments on the Quality of English Language

Major Revision 

Author Response

(The authors gave the same response as above.)

Round 2

Reviewer 1 Report

Comments and Suggestions for Authors

Dear editor

Following the authors’ detailed point-by-point response to all our comments and concerns, I am satisfied that they have adequately addressed the issues raised during the review process. Additionally, the authors have completed part of the supplementary analysis we suggested, and the remaining aspects have been appropriately deferred to future studies, as justified in their response.

In light of these revisions and clarifications, I find the manuscript significantly improved and more suitable for publication in your respected journal. Based on my assessment, I recommend that the manuscript be accepted for publication.

Best regards,

Reviewer 4 Report

Comments and Suggestions for Authors

None